# On the Generation of Digital Data and Models from Point Clouds: Application to a Pedestrian Bridge Structure

**F. Necati Catbas** [1,*] **, Jacob Anthony Cano** [2] **, Furkan Luleci** [1] **, Lori C. Walters** [3] **and Robert Michlowitz** [3]

1. Department of Civil, Environmental, and Construction Engineering, University of Central Florida, Orlando, FL 32816, USA; furkan.luleci@ucf.edu
2. Little Diversified Architectural Consulting, Charlotte, NC 28202, USA; jcano016@knights.ucf.edu
3. School of Modeling, Simulation and Training, University of Central Florida, Orlando, FL 32826, USA; lori.walters@ucf.edu (L.C.W.); robert.michlowitz@ucf.edu (R.M.)
*   Correspondence: catbas@ucf.edu

**Abstract:** This study investigates the capture of digital data and the development of models for structures with incomplete documentation and plans. LiDAR technology is utilized to obtain the point clouds of a pedestrian bridge structure. Two different point clouds with varying densities, (i) fine (11 collection locations) and (ii) coarse (4 collection locations), collected via terrestrial LiDAR, are analyzed to generate geometry and structural sections. This geometry is compared to the structural plans, which are then converted into numerical models (finite element—FE model) based on the point cloud data. Point cloud-based FE models (based on fine and coarse data) are compared with the structural plan-based FE model. It is observed that the static and dynamic responses are comparable within an acceptable range of a maximum difference of 5.5% for static deformation and an 8.23% frequency difference, with an average difference of less than 5%. Additionally, the dynamic properties of the fine and coarse point cloud FE models are compared with the operational modal analysis data obtained from the bridge. The fine and course point-cloud-based FE models, without model calibration, achieve an average accuracy of 8.76% and 9.94% for natural frequencies and a 0.89 modal assurance criterion value. The research found that the digital data generation yields promising results in this case for a bridge if documentation or plans are unavailable. With recent technologies and approaches such as digital twins, the connection between physical and virtual entities needs to be established by fusing digital models, sensorial information, and other data forms for better infrastructure management. Models such as those investigated and discussed in this paper can assist engineers with structural preservation in conjunction with monitoring data and utilization for digital twins.

**Keywords:** digital twin; point cloud; LiDAR; structural health monitoring; model updating; operational modal analysis

## 1. Introduction

Infrastructure preservation requires the integration of engineering, arts, and sciences, which are coordinated by a team consisting of architects, urban planners, and civil engineers with system domain knowledge. This paper summarizes a collaboration of engineers working in the area of health monitoring for structural preservation, a historian working in the area of historical preservation with virtual or augmented reality using LiDAR scans, and a computer scientist processing the large point cloud data of physical assets for simulation purposes. The focus of this paper is to investigate the extent to which it is possible to generate numerical models of structures with incomplete documentation and plans. Such models are expected to aid structural preservation in conjunction with monitoring data and be used in digital twinning, as well as for virtual entities for immersive visualization.

The monitoring, assessment, and evaluation of civil engineering structures have advanced over the last decade due to sustained global concern about the condition, performance, and preservation of infrastructure. An efficient structural assessment approach is necessary, especially due to the limited funds for preservation or re-building. Vast amounts of research and development are devoted to better understanding the conditions of civil structures, effective structural monitoring technologies for better maintenance, and management strategies [1]. Structural health monitoring (SHM) is expected to complement the traditional condition assessment techniques by providing objective data. SHM aims to track the responses collected from the structure via sensorial systems to identify the health status through damage diagnosis and prognosis for the subsequent decision-making phase [2–4]. SHM has been studied by many researchers over the last few decades. Many SHM-oriented applications and other auxiliary condition assessment and evaluation techniques have been proposed and tested on many case structures. Despite this trend, SHM has not yet become a routine practice, and is addressed in structural codes and standards.

Recently, there have been opportunities for and investments toward the digitization of the civil engineering industry while adopting sensor-based SHM applications. In this regard, several different technologies, such as Artificial Intelligence (AI) [5–7], Extended Reality (XR) [8], Building Information Management (BIM) [9], and digital twins (DTs) [10], are being investigated for the digitalization of the existent civil structure stock to enable more efficient structural asset sensing, control, and management approaches. While data analysis, visualization, and interpretation efficiency are leveraged through AI and XR technologies [11], BIM and DTs can be considered enabling technologies bringing heterogeneous data and knowledge to a more usable platform. As such, BIM is described as the digital representation of an existing civil asset to assist in design, construction, operation, and maintenance and form a reliable basis for decision-making, according to ISO 19650:2019.

On the other hand, DTs involve the employment of real-time sensorial data flow from the assets throughout their life cycle and leveraging data analytics via hybrid models (physics-based and data-driven models) to form what-if scenarios, future state predictions, and other parallel applications in addition to the use of BIM [12,13]. In this regard, BIM could be considered a more static asset representation, while DTs are more of a dynamic and evolving model throughout the model's lifespan since more sensor-based time-associated variables are engaged.

Studies [14,15] extended the three-dimensional DT model proposed in 2014 [16] to the five-dimensional DT model to satisfy the new requirements of the DT modeling, adding DT data and services. Figure 1 below shows the five dimensions of the DT model consisting of (1) a Physical Entity, such as a bridge equipped with various types of sensors collecting the real-time states (e.g., vibration, images) of the physical entity and working environment (e.g., temperature, humidity); (2) a Virtual Entity is a mirror image of the physical entity where it stands, for example, a CAD, numeric, and/or other type of numerical model of the bridge, providing the geometric and physical properties, as well as responding mechanisms and structural behaviors under different loading conditions and other external stressors; (3) Services, providing different services such as damage diagnostics services, model updating services, load rating estimation services, and/or remaining useful life prediction services to be used by other dimensions; (4) Digital Twin Data comprise the different kinds of data received from other dimensions and stored with a structured organization; (5) Connection ties the other dimensions together, forming the data threads using different communication technologies, e.g., LTE, LoRA, WiFi, radio, internet, etc.

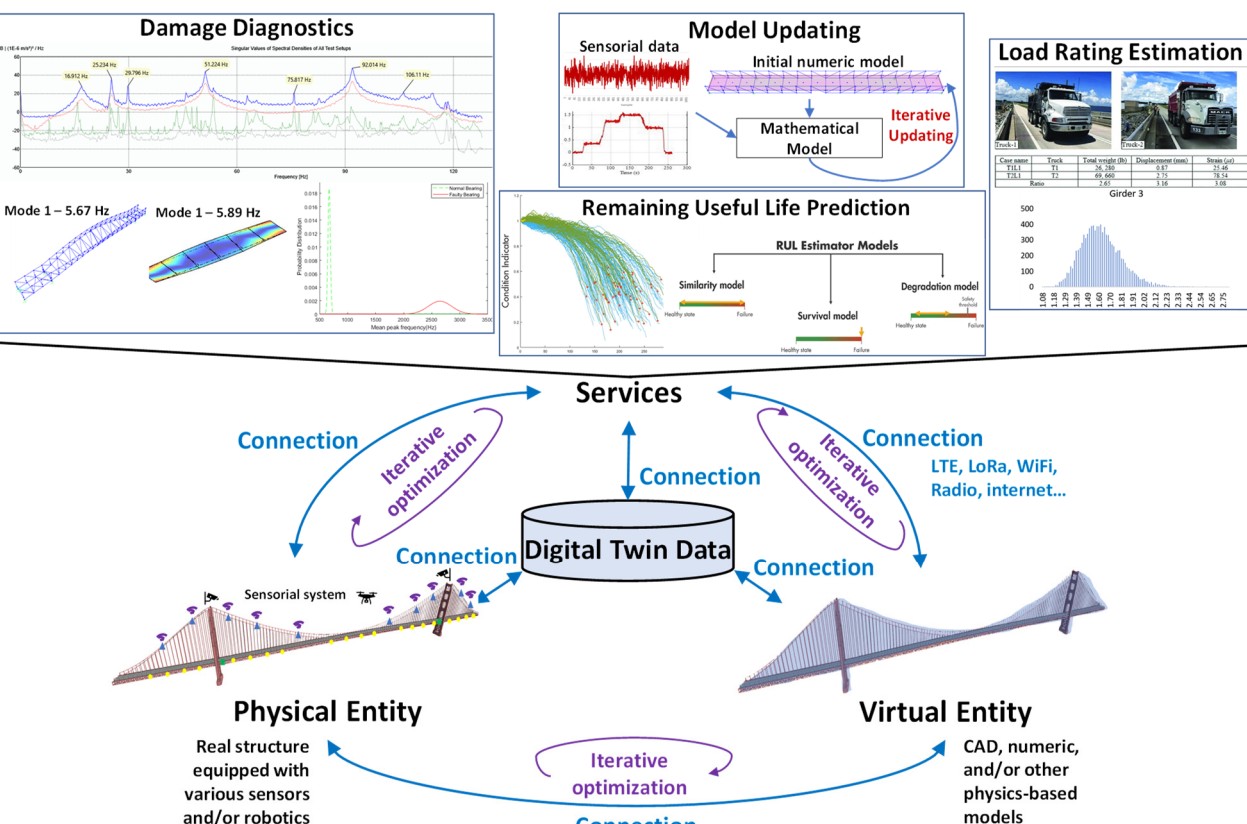

**Figure 1.** Five-dimension bridge digital twin concept model: Physical Entity, Virtual Entity, Services, Digital Twin Data, and Connection.

The digital twin (DT) concept was initially introduced to the public by Michael Grieves in 2002 at a Society of Manufacturing Engineers conference, where it was proposed as a model for product lifecycle management. Since then, it has also been referenced to under different terms, e.g., virtual twins, and later appeared in a report by NASA in 2010 as "Digital Twin" [17]. Over time, the three dimensions of DT (Physical Entity, Virtual Entity, and Connection) were extended to five dimensions to meet the technological needs, as explained above [15]. These five dimensions of DT could also be extended to further dimensions based on future needs.

One of the key elements of DT is the connection between physical and virtual entities, where the structural conditions of the existing structure (Physical Entity) are reflected in the physics-based numerical model (Virtual Entity) by fusing the sensorial information from the Physical Entity into the Virtual Entity in an iterative manner. This process is known as "model updating" or "model calibration" in the literature, where the aim is to update the initially built physics-based numerical model based on the information obtained from the monitoring and inspection data from the civil structure [18,19]. Many earlier works about model updating prior to the current popular subject, digital twins, can be found in the literature [2,20,21]. In addition, the applicability of the DT research outlooks and challenges, as well as DT system requirements, has been extensively studied in architecture, engineering, and construction [22–26].

Creating numerical models of existing civil structures is a critical component of the Virtual Entity dimension of a digital twin model, as shown in Figure 1. However, in order to build such a model, structural plans of the existing structures are needed, which might be occasionally unavailable in real-life scenarios. Even if they were available, the condition of the actual civil structure might not match what is depicted in the designs, which might have been damaged due to external stressors (e.g., earthquake or hurricane), constructed differently at the time of construction, or repaired using different structural elements. In

this case, generating a sufficiently accurate model of a physical asset without the design and construction plans and documents is important. Therefore, there is a need for alternative approaches for measuring and generating the geometry for a numerical model.

Among other methods such as RADAR, SONAR, and different types of cameras and 3D scanners (e.g., structured-light or time-of-flight 3D scanners), LiDAR (Light Detection and Ranging) (e.g., time-of-flight or phase LiDAR) emerges as one of the most precise and reliable technologies to scan and provide a 3D model of the existing civil structures [27–31].

In this regard, using LiDAR scans and point cloud data to create 3D models of existing structures may be a practical way to generate numerical models for developing a virtual entity as part of the digital twin. It is also worth noting that digital twins can be employed for the structural and historic preservation of civil infrastructure.

## 2. Objective and Scope

This study aims to study the feasibility of developing the point cloud from LiDAR scans for existing civil structures. It will also investigate how the structural responses of numerical models from in situ LiDAR scans compare to other models, as well as monitoring data. Based on the point-cloud-data-generated model, the geometric, static, and dynamic features of a pedestrian bridge located on the UCF campus are studied. Utilizing a 3D terrestrial laser scanner, the point cloud data of the pedestrian bridge structure are obtained and stitched (registered) together to create the 3D models to be imported into a Finite Element Analysis (FEA) program. A comparative analysis of the FEA results is carried out, comparing the models using the structural plans of the bridge versus the models using fine and coarse cloud densities. The structural displacements and reactions for static analysis and natural frequencies and mode shapes for the dynamic modal analysis are obtained via the FEA.

One type of global monitoring for tracking structural conditions is carried out by means of the dynamic signatures obtained using operational modal analysis. Here, the authors also explore whether models generated from non-contact, rapid, coarse-cloud-density LiDAR can provide reasonable dynamic signatures compared to actual data. The dynamic properties of three data sources (structural-plan-based/design FE model, point cloud-based FE models both fine and coarse) for the pedestrian bridge are also compared against experimental data using operational modal analysis (OMA), gathered in the field under golf cart excitation simulating operational vehicle-based excitation on highway bridges. The geometries of the structural-plan-based FE model and fine and coarse point cloud-based FE models are also examined and compared. The results indicate the viability of using point clouds and how the point cloud density influences the results. The study overview is illustrated in Figure 2.

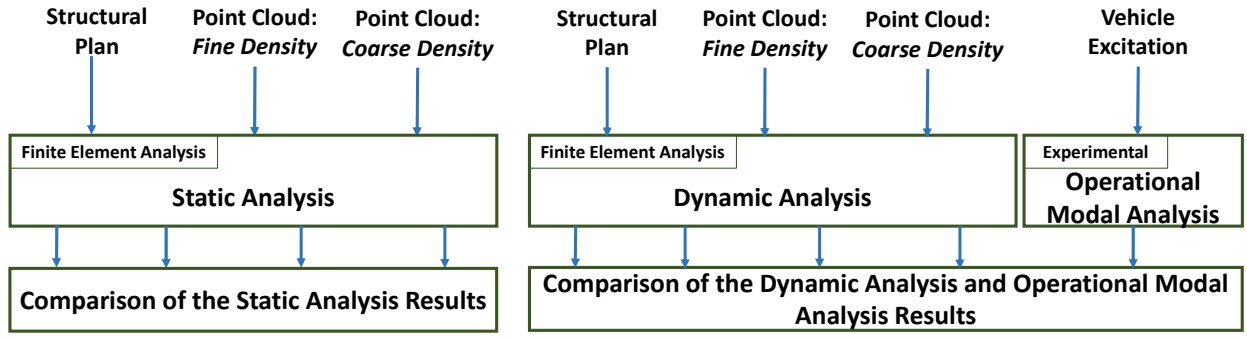

**Figure 2.** An overview of the study: comparison of the static analysis results of structural-plan-based (design) FE model and fine- and coarse-density point-cloud-based FE models of the pedestrian bridge structure; comparison of the dynamic analysis results with operational modal analysis results of structural-plan-based FE model and fine- and coarse-density point-cloud-based FE models of the pedestrian bridge structure.

### 3. Brief Background

Research on the viability of using point clouds as a tool for real-asset generation in 3D for structural maintenance, repair, and health monitoring applications of existing civil engineering structures has been widely explored in the literature [32–37]. The study [38] calculated the deformation accuracy of a structure using two point clouds, one with an undeformed shape and the other with a deformed shape. The results showed that the point cloud comparison produced a measurement accuracy of ±0.2 mm (95% confidence interval). Some other studies have been completed on the scanner's accuracy, its capability to obtain real-life measurements, and what factors affect the accuracy of the results.

Another study [39] describes how decisions made throughout the "registration" of a point cloud impact the accuracy of the point cloud in terms of the dimensions of the point cloud it yields. The research by [40] demonstrated the percentage differences obtained when the photogrammetry method was compared to typical hand measurements and structural plans. The study showed that the photogrammetry technique only varied from 0.06 to 1.43% compared to the hand measurement method and 0.23 to 8.00% compared to structural plans.

Furthermore, one study [41] investigated how LiDAR can be used to estimate smaller cross-sectional dimensions of a functioning bridge and quantify the errors observed in the capacity calculations, rather than just simple percentage errors. The authors of that study conducted 16 LiDAR scans of an 11-span steel girder bridge while it was in normal operation. They extracted various dimensional measurements from the data, both directly and using conventional plane-fitting methods (Plane Fitting and RANSAC). The findings showed that the dimensions obtained through Plane Fitting led to flexural capacities that were 4–7% lower than those computed using the bridge plans' dimensions. The RANSAC method estimated errors within the range of 7–10%, whereas the dimensions obtained directly from the point cloud data resulted in capacity errors of 9–13%. Throughout the study, all dimensions were estimated conservatively due to potential sources of errors. However, it was noted that the distortion of elements caused by fabrication stresses could lead to an overestimation of the dimensions if assumptions of the planar surfaces were made.

Another study [42] introduced a novel algorithm that utilizes LIDAR data to automatically and separately measure the deformations caused by torsional and bending loading in beams, especially in open cross sections like I, U, and L shapes, which have low torsional strength. These deformation values were then used to calculate shear stresses. The methodology was tested on a physical beam in a laboratory under various loads, and the algorithm's results were compared with direct measurements and 3D FEA simulations. According to the authors, the new algorithm's results aligned well with the other methods, confirming its validity for monitoring metal structures' health.

M. Song et al. [43] examined finite element model updating to identify induced damage in a two-story concrete masonry-infilled building. Vibration data and LiDAR scans were utilized. The test building structure experienced significant damage from an earthquake and was slated for demolition after various acceleration measurement tests. Damage was introduced by removing the exterior walls during forced tests. The modal parameters were estimated using ambient and forced vibration measurements for reference and damaged states. LiDAR data were employed to detect surface defects and changes due to the wall removal and the tests. Initial FE models, tuned and untuned, were created based on the inspections and tests. Reference models were calibrated to represent the structure's states using these initial models. The reference models were then updated to match the measured data at the damaged state, assessing damage via stiffness reduction. The estimated damage closely aligns with the nominal values and LiDAR results, demonstrating agreement between the untuned and tuned models.

The research in [44] pointed out that traditional monitoring of bridge response using discrete sensors can be time-consuming and offer limited insights. Thus, the author studied the application of LiDAR to analyze how bridges deflect in response to varying live and dead loads. The study involved monitoring two bridges under the weight of a loaded

triaxial truck and two others during phased construction with concrete deck pouring. All four bridges, representing diverse types, were assessed using LiDAR, which accurately captured high-fidelity and full-field deflection shapes. Specifically, a numerical model was created using established parameters for an inverted tee girder bridge measuring 19.81 m. However, the traditional FEA failed to predict the bridge's response to live loads accurately. This disparity in uniform displacement was revealed using the LiDAR point clouds. This case underscores LiDAR's advantages in providing full-field response data and simplifying the load testing process.

As described in the literature review presented briefly, LiDAR has a wide variety of uses in examining structure and infrastructure systems. Nevertheless, there is still a large amount of room in exploring the use of LiDAR for assessing and evaluating existing civil structures, particularly using LiDAR for structural analysis, such as for static and dynamic analysis tests, which the study herein examines on two different case structures.

## 4. Case Structure

The pedestrian bridge (Figure 3) is on Gemini Boulevard North on the UCF campus. This pedestrian bridge structure is a steel truss system that forms both vertical sides and the horizontal side carrying the reinforced concrete deck. It is 177 feet long by 12 feet wide and has three spans. The bridge primarily sustains pedestrian and small utility vehicle loads. The essential elements of this bridge comprise steel HSS and W-sections. More details of the pedestrian bridge can be found in these studies [45].

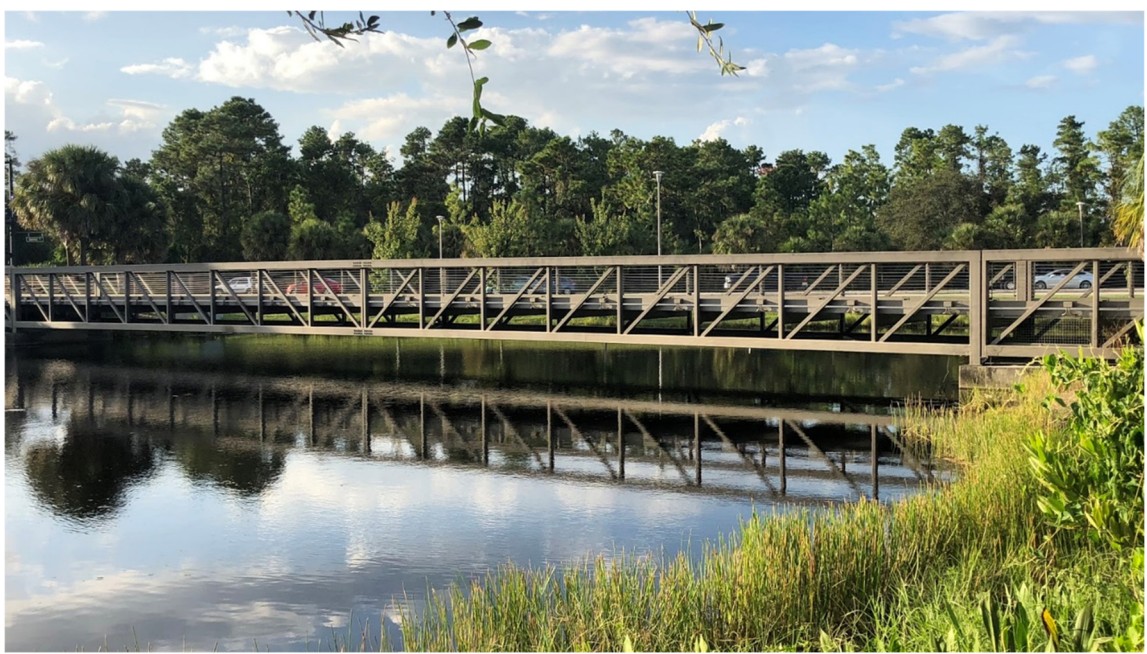

**Figure 3.** The pedestrian bridge structure.

## 5. Methodology

### 5.1. Data Collection and Processing

The pedestrian bridge was scanned using a Leica ScanStation P40 3D laser scanner to obtain its point clouds. It took 5 hours and 45 minutes to complete the 11 scans. On the bridge, three 4-inch black-and-white artificial circular targets with the labels 1, 2, and 3 were placed to create triangular coordination with the targets. Targets were manually rangefound and marked on the scanner's screen following the initial scan. After each scan, the scanner must always capture consistently named targets, permitting triangulation with at least two targets per scan. All three targets are marked for certain scans when connecting scans that cannot "see" the same two targets as the previous scans.

The scanner was moved to two additional locations on the north side of the bridge to gather data from that side. Before each scan, the device must be leveled and checked to be in sight of two targets. Once each scan is complete, the target distances must be captured. Typically, only two targets are ranged instead of three, which will only occur when connecting scans where a previously used target cannot be sighted.

The scanner is then repositioned for four scans on the south side, one scan at the west end of the bridge, and one scan at the east end. For every scan, the scanner must be moved to a new location and leveled, its range set to include at least two targets, and the targets captured. The scanner is finally moved to two different spots on the bridge. The target one is moved to the north of the bridge prior to the initial scan on the bridge. If one of the original targets is kept in place and used as a reference for the newly moved target, moving a target is possible. Because the capture range is set to 360 degrees for these two scans, manually dictating an angle wedge is unnecessary. Figure 4 shows the aerial view of the scanning locations and corresponding photos taken at roughly equal intervals.

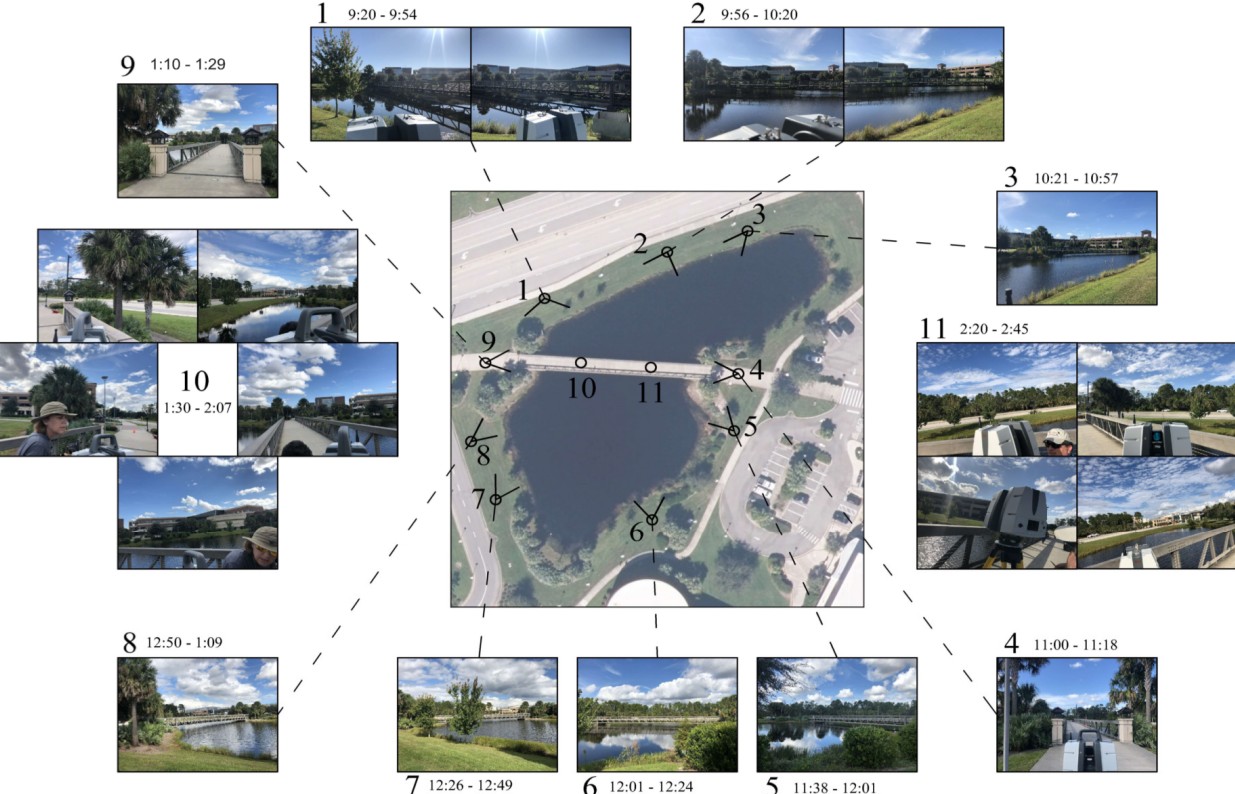

**Figure 4.** Aerial view of the scanning locations of the pedestrian bridge.

Leica Cyclone REGISTER 360, point cloud registration software, was utilized to join the 11 scans to produce a single point cloud. Users can adjust, edit, and "stitch" together scans using a semi-automated process, after which a registration report is generated, providing scan metrics to evaluate the project. In the case of this study, the program used coordinate triangulation to combine the 11 scans into a single encompassing point cloud using the target data collected on site for each setup. Target use is noted to reduce the software editing time by roughly 75% while lengthening on-site scanning times and reducing cloud-to-cloud errors. The placement of the scanner is shown in Figure 5a, according to the registration report from Cyclone. The strong connections between the scanning locations (yellow triangles) shown in the figure's green lines allowed for scanning overlap and made it easier to stitch the various scans together while outputting an overall scan without holes.

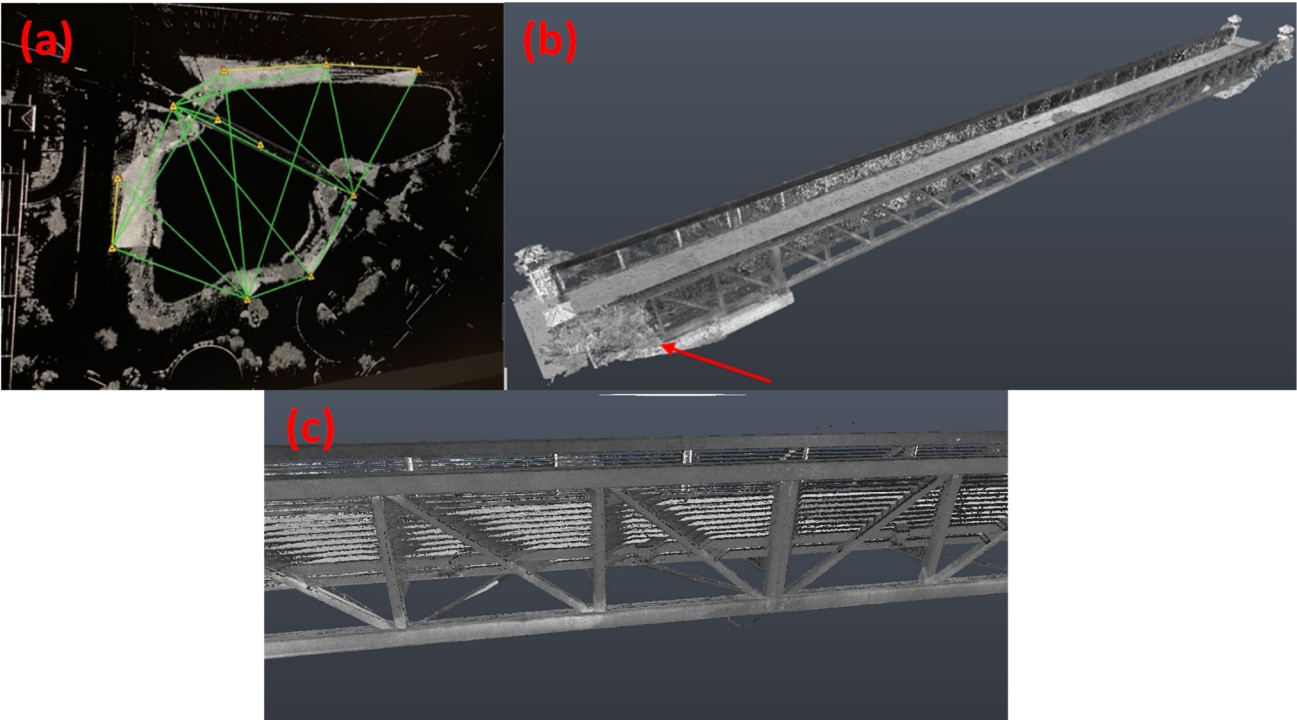

**Figure 5.** (**a**) Scan locations diagram, (**b**) fine point cloud after the registration (red arrow showing the leftover shrubbery), (**c**) zoomed-in fine point cloud of the pedestrian bridge.

The 11 scans yielded data points from surfaces outside the bridge, resulting in large file sizes of more than 24 gigabytes. This over-scanning results from the laser scanner's capability to measure surfaces at a distance of up to 270 m. A single scan can contain anywhere from 50 to 190 million points. In Cyclone, the unnecessary and distracting points were eliminated, for example, trees, shrubbery, cars, buildings, people, pond reflections, and black-and-white targets. The complete removal of the shrubbery from the point cloud is not plausible, as shown in Figure 5b, due to the quantity of shrubbery at the site and its proximity to the bridge. The point cloud was registered in less than two hours overall. Figure 5c shows a zoomed-in image demonstrating the density of the fine point cloud. Such point clouds can also be employed in virtual reality tours of the bridge, which the authors will discuss in a future paper.

The edited point cloud is opened using Autodesk ReCap after the registration process. ReCap was selected because it can open the file format that Cyclone REGISTER 360 uses. ReCap can also enable and disable scans, which impacts the point cloud's density. When all 11 scans are active, scanning the fine point cloud on site will take 5 h and 45 min. The coarse point cloud, which would have taken 1 h and 36 min to scan on site, was created by turning off seven scans and leaving four scans on for a total of 11. The coarse scan includes locations 3, 4, 5, and 8. The defining tool that enables a user to establish three levels of point cloud density for comparison is ReCap's ability to turn scans on and off.

After selecting the proper density level, the file is saved as an RCP file to be compatible with Autodesk Inventor. Sections are rendered in Inventor using the point cloud data imported from ReCap. To the best of their ability, the Inventor user shall estimate an accurate section size for members of the pedestrian bridge using the point cloud visual as a base. Once the point cloud visual is imported into Inventor, the following steps are implemented to render sections onto the point cloud visual within Inventor for the fine and coarse levels of point cloud density: (1) Add the ReCap file to an "Assembly" file in Inventor; (2) Construct a "Part" within the Assembly to draw a 2D sketch on the point cloud's sides; (3) Before beginning the 2D sketch, place a "Work Plane" on a flat surface of the user's choosing; (4) Align the sketch lines with the visible sections to produce a

center-to-center sketch using the point cloud as a guide; (5) After the sketch is finished, add frames and offset them appropriately to match the sections visible in the point cloud as closely as possible (users will need to use a trial-and-error method to find the section size that they believe most closely resembles the shape seen in the point cloud); (6) Repeat the procedure for all relevant sides and sections of the structure.

The sketches are imported into AutoCAD after the sections have been decided upon, and the point cloud has a fully rendered representation in Inventor. Since AutoCAD DXF files are compatible with the FEA package of choice (SAP2000), this step is required to input the sketch into the software. The sketches are properly aligned in AutoCAD once they have been imported, ensuring that the frame lines are connected, and there are no alignment errors. Since the user's judgment solely determines the member sizes and length values, they are completely up to the user's discretion. Assume, for instance, that an AutoCAD sketch has a length of 1200.34 inches. The user can infer that the member line is 1200 inches long. After AutoCAD has finished this process, the DXF file is then imported into SAP2000 for FEA. For more details about the TLS data collection parameters, the readers are referred to a study by Cano [46].

*5.2. Comparison of the Static Analysis Results*

In order to determine whether there is a relationship between a point cloud's density and the accuracy of its results, the outputs of both point cloud models are compared to those of the structural plan (design) model. When both point clouds were dimensioned alongside the structural plans, they showed to be extremely accurate in terms of length, width, height, and spacing. All point cloud cases have an accuracy that is consistently near or above 99% in all three directions, and all are within 1 inch of the structural plan cases. According to the on-site measurements taken at the pedestrian bridge via a laser meter, the structural plans and the as-built structure had no more than a 1-inch dimension difference. With this in mind, the point clouds attempted to match the structural plan (design) model as closely as possible. The result is the as-built structure, which the structural plan model represents. As a result, the point cloud model comparison uses the pedestrian bridge's structural plans as a guide.

A pedestrian bridge model is made using Autodesk Inventor based on the structural plans. For the HSS sections, ASTM A500 is used for the wide flange beams, and ASTM A572 is used. The structural plans' materials list is used to define the section properties. These sections consist of HSS10x10x3/8 for the top and bottom chords, HSS6x4x3/8 for the vertical and splice vertical members, HSS10x10x3/8 for the end vertical members of spans 1 and 3, HSS10x4x3/8 for the diagonal members, HSS4x4x1/4 for the diagonal brace members, and W12x22 for the floor beams and splice floor beams.

Distributed loads are applied to floor beams and splice floor beams to represent the concrete deck, which has a minimum compressive strength of 4000 psi and a maximum weight of 145 pcf. The calculated distributed loads are based on a 5-inch deck and an 8-foot-wide tributary. Only the dead load is taken into account for the dynamic modal analysis. As the structural plans state, the floor beams are given an additional 100 psf for the static analysis. Figure 6 displays the SAP2000 model for a pedestrian bridge along with the base joint numbers designated for comparing the base joint reaction values. Because the bridge is symmetrical and the opposite end span produces the same reaction forces, only four reactions are depicted.

SAP2000 is used to perform a static analysis of the bridge using the dimensions and sections listed in the structural plans. The structural plans subject the structure to dead and live loads, and the midpoints of the center span experienced the greatest deformation with a value of $-3.66$ inches in the *z*-axis (vertical to the span axis). Tables 1 and 2 compare the midpoint displacement results and base joint reactions of the structural-plan-based (design) FE model to the displacements and reactions obtained from the fine and coarse point-cloud-based FE models, respectively.

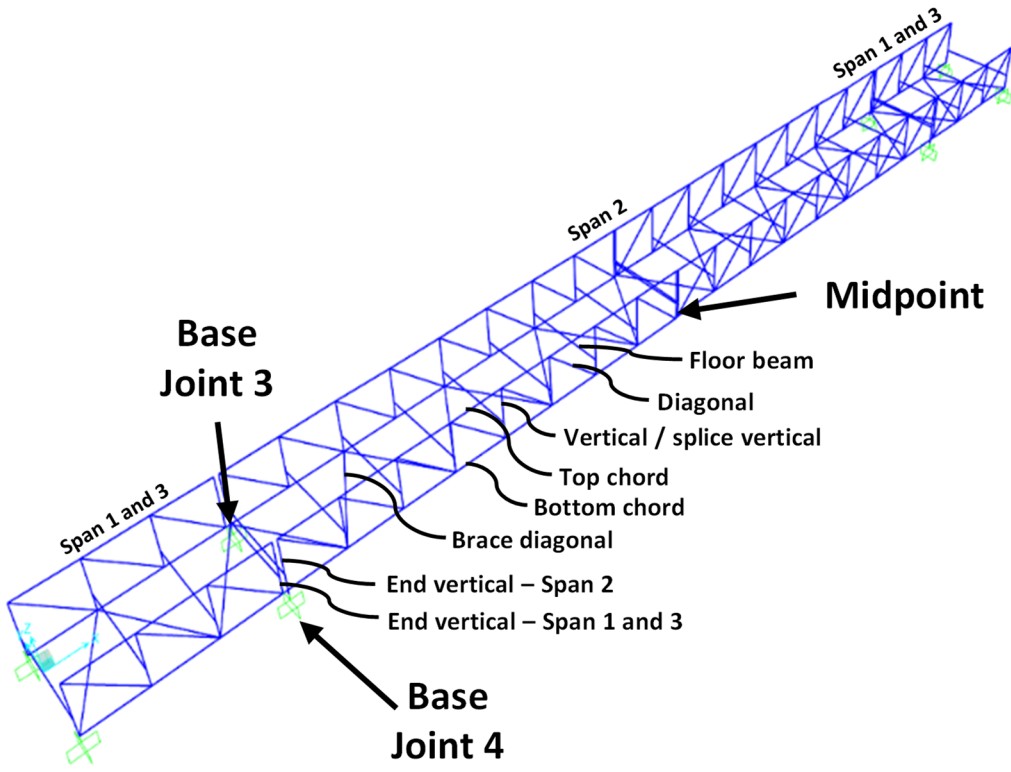

**Figure 6.** An image of the pedestrian bridge model in SAP2000 shows the base joint numbers for comparing the base joint reaction values. Note that the member names were taken as is from the structural plans of the bridge.

**Table 1.** Midpoint displacement comparison results between structural-plan-based (design) FE model and fine and coarse point-cloud-based FE models.

| Midpoint Displacement (Inches) | | | | | | |
|---|---|---|---|---|---|---|
| Location | Structural-Plan-Based FE Model | Fine Point-Cloud-Based FE Model | % Difference | Structural-Plan-Based FE Model | Coarse Point-Cloud-Based FE Model | % Difference |
| Midpoint of Center Span | −3.66 | −3.53 | 3.70 | −3.66 | −3.47 | 5.50 |

**Table 2.** Base joint reaction comparison results between structural-plan-based (design) FE model and fine and coarse point-cloud-based FE models.

| Base Joint Reactions (Kip) | | | | | | |
|---|---|---|---|---|---|---|
| Base Joint No | Structural-Plan-Based FE Model | Fine Point-Cloud-Based FE Model | % Difference | Structural-Plan-Based FE Model | Coarse Point-Cloud-Based FE Model | % Difference |
| 3 | 79.47 | 79.38 | 0.11 | 79.47 | 79.41 | 0.08 |
| 4 | 79.49 | 79.46 | 0.04 | 79.49 | 79.47 | 0.03 |

The bridge underwent static analysis in SAP2000 based on the dimensions and sections rendered after the model generation procedure implemented for two different point cloud density levels (fine and coarse). The structural plans stipulate that the structure is subject to dead and live loads. Every point cloud model case's midpoint experienced the greatest deformation, albeit at varying intensities. The maximum displacements for the fine and coarse point clouds are −3.53 inches and −3.47 inches. The midpoint displacement results, as well as the base joint reactions of the fine and coarse point-cloud-based FE models, can

be seen in Tables 1 and 2 in a comparative fashion to the displacements and reactions obtained from the structural-plan-based (design) FE model.

Comparing the displacement results of the fine and coarse point-cloud-based FE models to the structural-plan-based FE model shows the percentage differences in Table 1. The maximum deflection of the fine point cloud is 3.7% less than the actual deformation value, which is not far off. However, the coarse point cloud only deviates by 5.5%, placing it close to the structural plans' stated deflection value. In this regard, it is clear that the fine point cloud exhibits the most accurate densities. Table 2 displays the percentage variations between each point cloud and the structural plans for base joint reactions. According to the results of the fine and coarse point-cloud-based FE models, the two base joints are close to zero, and the others are in the high 21.85–21.87% range.

*5.3. Comparison of the Dynamic Analysis Results*

The structural plans of the pedestrian bridge are subjected to dynamic analysis in SAP2000. Since the bridge's dynamic motion is no longer restricted to the end spans after the eighth mode, a total of nine modes are examined. This is significant because, as shown in the following figures, the center span is the only place where the first eight modes can move. The nine modes for the fine and coarse FE models based on point clouds also produce the same mode shapes as the FE model based on structural plans. Due to the different structural member dimensions in each FE model, the frequency and load participation values differ, though not significantly.

As shown in Table 3, when the dynamic analysis outcomes from the fine point-cloud-based FE model are contrasted with those from the structural-plan-based (design) FE model, there are percentage differences that can be attributed to the various member sizes rendered. This study details the pedestrian bridge's structural design and highlights member size changes using red [45]. It is evident that while obtaining the correct width and height for HSS members is simple, matching the correct thickness is much more challenging. The three others are below 30%, with the largest cross-sectional area percentage difference being 32.10%. The members provided in the structural plans match half of the members rendered. The average cross-sectional area difference is just under 12% when all the structure members are considered.

**Table 3.** Member size comparison: structural-plan-based (design) FE model vs. fine point-cloud-based FE model.

| Location | Structural-Plan-Based FE Model | Fine Point-Cloud-Based FE Model | Cross-Sectional Area % Difference |
|---|---|---|---|
| Top Chord | HSS 10x10x3/8 | HSS 10x10x3/8 | 0.00% |
| Bottom Chord | HSS 10x10x3/8 | HSS 10x10x3/8 | 0.00% |
| Vertical/Splice Vertical | HSS 6x4x3/8 | HSS 6x4x5/16 | 16.10% |
| End Vertical—Span 2 | HSS 10x10x3/8 | HSS 10x10x1/2 | 26.50% |
| End Vertical—Spans 1 and 3 | HSS 10x4x3/8 | HSS 10x4x3/8 | 0.00% |
| Diagonal | HSS 4x4x1/4 | HSS 4x4x5/16 | 19.40% |
| Brace Diagonal | HSS 3x3x1/4 | HSS 3x3x3/8 | 32.10% |
| Floor Beam | W12x22 | W12x22 | 0.00% |
| | | **Average =** | **11.76%** |

Surprisingly, compared to the two earlier point-cloud-based FE models, the coarse point-cloud-based model produces a better comparison with the member size given in structural plans. Table 4 reveals the variations in member sizes. As is evident, all member sizes, with the exception of two, matched those found in the structural plans. The two members have a 16.70% and a 21.60% difference. The coarse point cloud is more accurate regarding member sizes because it has the highest number of matching members, which reduces the average cross-sectional area difference to under 5%.

**Table 4.** Member size comparison: Structural-plan-based (design) FE model vs. coarse point-cloud-based FE model.

| Location | Structural-Plan-Based FE Model | Coarse Point-Cloud-Based FE Model | Cross-Sectional Area % Difference |
|---|---|---|---|
| Top Chord | HSS 10x10x3/8 | HSS 10x10x3/8 | 0.00% |
| Bottom Chord | HSS 10x10x3/8 | HSS 10x10x3/8 | 0.00% |
| Vertical/Splice Vertical | HSS 6x4x3/8 | HSS 6x4x3/8 | 0.00% |
| End Vertical—Span 2 | HSS 10x10x3/8 | HSS 10x10x3/8 | 0.00% |
| End Vertical—Spans 1 and 3 | HSS 10x4x3/8 | HSS 10x4x5/16 | 16.70% |
| Diagonal | HSS 4x4x1/4 | HSS 4x4x3/8 | 21.60% |
| Brace Diagonal | HSS 3x3x1/4 | HSS 3x3x1/4 | 0.00% |
| Floor Beam | W12x22 | W12x22 | 0.00% |
| | | **Average =** | **4.79%** |

One of the reasons that can be attributed to the effectiveness of the coarse data set over the fine data set is that only four locations that well cover the bridge with few obstacles provide data for the coarse scan. As can be seen in Figure 3, locations 3, 4, 5, and 8 have very clear fields of view with minimum interference, well covering the bridge from both sides as well from the top. The fine scan has additional data that may have some uncertainty due to the blocked sections of the bridge, where the obtained point cloud includes uncertainties and artifacts when used to obtain the sections.

Table 5 compares the natural frequencies between the fine and coarse point-cloud-based FE models and the structural-plan-based (design) FE model. The fine point cloud member sizes differ from those specified by the structural plans. The percentage differences for the fine point cloud comparison case range from 0.18 to 8.23%. The values of the point-cloud-based FE models have higher frequencies than those of the structural-plan-based FE models (except the final mode). The percentage differences in the coarse point-cloud-based FE model comparison range from 0.13% to 4.66%. The accuracy of both point clouds is highest for this range because it is the smallest. Despite being the most accurate overall, it is slightly less accurate than the fine cloud-based FE model in three modes (Modes 2, 5, and 9).

**Table 5.** Comparing natural frequencies of structural-plan-based (design) FE model to fine and coarse point-cloud-based FE model.

| | Frequency (Hz) | | | | | |
|---|---|---|---|---|---|---|
| Mode | Structural-Plan-Based FE Model | Fine Point-Cloud-Based FE Model | % Difference | Structural-Plan-Based FE Model | Coarse Point-Cloud-Based FE Model | % Difference |
| 1 | 2.15 | 2.23 | 3.59 | 2.15 | 2.16 | 0.46 |
| 2 | 2.54 | 2.58 | 1.55 | 2.54 | 2.6 | 2.31 |
| 3 | 3.68 | 4.01 | 8.23 | 3.68 | 3.71 | 0.81 |
| 4 | 4.92 | 5.24 | 6.11 | 4.92 | 4.93 | 0.20 |
| 5 | 6.95 | 7.13 | 2.52 | 6.95 | 7.29 | 4.66 |
| 6 | 7.59 | 8.04 | 5.60 | 7.59 | 7.58 | 0.13 |
| 7 | 8.45 | 8.95 | 5.59 | 8.45 | 8.81 | 4.09 |
| 8 | 9.59 | 10.01 | 4.20 | 9.59 | 9.56 | 0.31 |
| 9 | 11.02 | 11 | 0.18 | 11.02 | 10.72 | 2.80 |

It is also seen in Table 5 that the coarse point-cloud-based model provided closer approximations to the structural-plan-based design model. The average errors of the fine and coarse point-cloud-based models are 4.17% and 1.75%, respectively. However, one can also consider that both models provide reasonably good approximations with a maximum single-mode difference of less than 10%.

The load participation factor provides insight into identifying and assessing the influence of different loads on the structural response. It helps engineers make informed decisions during the design and optimization phases, especially ensuring that structures can withstand dynamic forces and environmental conditions. Table 6 compares the modal load participation factors of the fine point-cloud-based FE model and the structural-plan-based (design) FE model. The percentage difference is regarded as the difference between the two because the values are already expressed as percentages. As can be seen, all three directions of the load participation factor results are somewhat close together, with the biggest difference barely exceeding 11.92%. Comparing the load participation factors between the coarse point-cloud-based FE model and the structural-plan-based (design) FE model is shown in Table 7. The coarse point-cloud-based FE model produces larger differences in the load participation factors, with 23.08% found to be the biggest difference. Both models provide a reasonable comparison with the structural-plan-based (design) FE model, especially in the vertical direction ($z$-direction) where the main load carrying capacity due to the static and dynamic loads is to be considered.

**Table 6.** Load participation comparison: structural-plan-based (design) FE model vs. fine point-cloud-based FE model.

| Direction | Static (%) | | | Dynamic (%) | | |
|---|---|---|---|---|---|---|
| | Structural-Plan-Based FE Model | Fine Point-Cloud-Based FE Model | % Difference | Structural-Plan-Based FE Model | Fine Point-Cloud-Based FE Model | % Difference |
| $x$-axis | 10.37 | 11.31 | 8.31 | 1.70 | 1.93 | 11.92 |
| $y$-axis | 99.55 | 99.37 | 0.18 | 82.63 | 80.62 | 2.49 |
| $z$-axis | 99.15 | 99.2 | 0.05 | 56.46 | 56.27 | 0.34 |

**Table 7.** Load participation comparison: structural-plan-based (design) FE model vs. coarse point-cloud-based FE model.

| Direction | Static (%) | | | Dynamic (%) | | |
|---|---|---|---|---|---|---|
| | Structural-Plan-Based FE Model | Coarse Point-Cloud-Based FE Model | % Difference | Structural-Plan-Based FE Model | Coarse Point-Cloud-Based FE Model | % Difference |
| $x$-axis | 10.37 | 12.25 | 15.35 | 1.70 | 2.21 | 23.08 |
| $y$-axis | 99.55 | 99.48 | 0.07 | 82.63 | 81.69 | 1.15 |
| $z$-axis | 99.15 | 99.24 | 0.09 | 56.46 | 56.55 | 0.16 |

*5.4. Comparison of Dynamic Analysis with Operational Modal Analysis Results*

Operational modal analysis (OMA) is a typical ambient dynamic monitoring method to assess structural performance or change when tracked over time. OMA provides dynamic properties, mainly frequencies and mode shapes, which are related to the structural stiffness and boundary conditions of a particular structure. Here, OMA is used to compare the structural-plan-based (design) FE model and the fine and coarse point-cloud-based FE models against the experimental data obtained from the pedestrian footbridge. The first five modes are retrieved from the study and obtained from the experimental data analysis. The dynamic modal analysis completed for 9 modes must be increased to 18 to match the five modes of OMA since more modes are obtained due to the spatial resolution of the numerical models. As a result, the final matrix dimension for each comparison becomes 18x5 when using the modal assurance criterion (MAC) to compare the modes of the structural-plan-based (design) FE model and the fine and coarse point-cloud-based FE models with those from OMA.

The MAC values of the OMA modes compared to the modes of the structural-plan-based (design) FE model are displayed in Table 8a. The nine significant MAC values are bolded in the table, as can be seen. Five mode correlations are greater than 90%, three

are greater than 80%, and one is greater than 70%. The second mode discovered by the OMA, nearly identical to the third mode discovered by the structural plans, is represented by the highest value of 97.4%. The MAC values of the modes of OMA compared to the modes of fine point-cloud-based FE models are displayed in Table 8b. The FE model's fine point-cloud-based MAC results produce 10 significant values, with 5 of the 10 values above 90%, 3 above 80%, and the final 2 above 70%. The fine point-cloud-based FE model exhibits similar results with regard to the significance values of the structural plan MAC. This similarity in outcomes shows how accurate point cloud data can be compared to on-site results, even when those results are from structural plans.

**Table 8.** (a) MAC values of the modes of OMA versus modes of structural-plan-based (design) FE model; (b) MAC values of the modes of OMA versus modes of fine point-cloud-based FE models; (c) MAC values of the modes of OMA versus modes of coarse point-cloud-based FE models.

| | | OMA Mode No | | | | | | | OMA Mode No | | | | | | | OMA Mode No | | | | |
|---|---|---|---|---|---|---|---|---|---|---|---|---|---|---|---|---|---|---|---|---|
| **(a)** | 1 | 2 | 3 | 4 | 5 | | **(b)** | 1 | 2 | 3 | 4 | 5 | | **(c)** | 1 | 2 | 3 | 4 | 5 |
| 1 | 0.00 | **0.97** | **0.87** | 0.00 | 0.00 | | 1 | 0.00 | **0.97** | **0.87** | 0.00 | 0.00 | | 1 | 0.00 | **0.97** | **0.87** | 0.00 | 0.00 |
| 2 | **0.95** | 0.00 | 0.10 | 0.01 | 0.00 | | 2 | **0.95** | 0.00 | 0.10 | 0.01 | 0.00 | | 2 | **0.95** | 0.00 | 0.10 | 0.01 | 0.00 |
| 3 | 0.01 | **0.97** | **0.88** | 0.00 | 0.00 | | 3 | 0.01 | **0.98** | **0.88** | 0.00 | 0.00 | | 3 | 0.01 | **0.97** | **0.88** | 0.00 | 0.00 |
| 4 | 0.00 | 0.00 | 0.00 | 0.00 | 0.00 | | 4 | 0.00 | 0.00 | 0.00 | 0.00 | 0.00 | | 4 | 0.00 | 0.00 | 0.00 | 0.00 | 0.00 |
| 5 | 0.00 | 0.00 | 0.00 | **0.92** | 0.03 | | 5 | 0.00 | 0.00 | 0.00 | **0.92** | 0.03 | | 5 | 0.00 | 0.00 | 0.00 | **0.92** | 0.03 |
| 6 | 0.00 | 0.08 | 0.08 | 0.00 | 0.00 | | 6 | 0.00 | 0.07 | 0.07 | 0.00 | 0.00 | | 6 | 0.00 | 0.08 | 0.08 | 0.00 | 0.00 |
| 7 | 0.00 | 0.00 | 0.00 | 0.00 | 0.00 | | 7 | 0.00 | 0.00 | 0.00 | 0.00 | 0.00 | | 7 | 0.00 | 0.00 | 0.00 | 0.00 | 0.00 |
| 8 | 0.00 | 0.00 | 0.00 | 0.00 | 0.00 | | 8 | 0.00 | 0.00 | 0.00 | 0.00 | 0.00 | | 8 | 0.00 | 0.00 | 0.00 | 0.00 | 0.00 |
| 9 | 0.00 | 0.07 | 0.08 | 0.00 | 0.00 | | 9 | 0.00 | 0.24 | 0.23 | 0.00 | 0.00 | | 9 | 0.00 | 0.38 | 0.36 | 0.00 | 0.00 |
| 10 | 0.00 | 0.00 | 0.00 | 0.01 | 0.00 | | 10 | 0.00 | 0.00 | 0.00 | 0.00 | 0.00 | | 10 | 0.00 | 0.00 | 0.00 | 0.00 | 0.00 |
| 11 | 0.00 | 0.01 | 0.01 | 0.00 | 0.01 | | 11 | 0.03 | 0.02 | 0.03 | 0.00 | 0.11 | | 11 | 0.00 | 0.00 | 0.01 | 0.00 | 0.01 |
| 12 | 0.00 | 0.00 | 0.00 | 0.00 | 0.01 | | 12 | 0.00 | 0.00 | 0.01 | 0.00 | 0.01 | | 12 | 0.00 | 0.00 | 0.00 | 0.00 | 0.00 |
| 13 | 0.08 | 0.00 | 0.01 | 0.00 | **0.74** | | 13 | 0.00 | 0.00 | 0.00 | **0.75** | 0.03 | | 13 | 0.00 | 0.00 | 0.00 | 0.01 | 0.00 |
| 14 | 0.00 | 0.00 | 0.00 | 0.01 | 0.00 | | 14 | 0.00 | **0.96** | **0.88** | 0.00 | 0.00 | | 14 | 0.00 | 0.04 | 0.04 | 0.00 | 0.03 |
| 15 | 0.00 | 0.00 | 0.00 | 0.35 | 0.01 | | 15 | 0.00 | 0.00 | 0.00 | 0.00 | 0.00 | | 15 | 0.00 | 0.00 | 0.00 | 0.04 | 0.00 |
| 16 | 0.01 | 0.31 | 0.25 | 0.00 | 0.11 | | 16 | 0.00 | 0.57 | 0.52 | 0.00 | 0.05 | | 16 | 0.00 | 0.00 | 0.00 | 0.00 | 0.00 |
| 17 | 0.00 | 0.00 | 0.00 | 0.00 | 0.00 | | 17 | 0.00 | 0.00 | 0.00 | 0.01 | 0.00 | | 17 | 0.00 | **0.97** | **0.88** | 0.00 | 0.00 |
| 18 | 0.01 | **0.94** | **0.86** | 0.00 | 0.00 | | 18 | 0.08 | 0.00 | 0.01 | 0.00 | **0.74** | | 18 | 0.08 | 0.00 | 0.01 | 0.00 | **0.75** |

*Row labels for columns: Structural-Plan-Based FE Model Mode No (a); Fine Point-Cloud-Based FE Model Mode No (b); Coarse Point-Cloud-Based FE Model Mode No (c).*

The MAC values of the OMA modes compared to the modes of the coarse point-cloud-based FE model are shown in Table 8c. Nine significant values are bolded in the table: five above 90%, three above 80%, and one above 70%. These percentages, including the highest correspondence value of 97.4%, agree with the structural-plan-based (design) FE model results. This value is obtained by comparing the second mode of the OMA with the third mode of the coarse point cloud. This comparison follows the same pattern as the fine point-cloud-based and structural-plan-based FE models.

Table 9 contrasts the modes of the OMA with the modes of the structural-plan-based (design) FE model. The table gives the respective MAC value for the mentioned mode comparison, along with the percentage differences for frequencies. The corresponding structural-plan-based FE model's mode is selected for the frequency comparison by selecting the highest MAC value for every OMA mode. The table demonstrates how closely the structural-plan-based FE model adheres to the frequency identified by the OMA data. The comparison between the structural plan Mode 3 and the OMA Mode 3 is the only exception (highlighted in red in the table), resulting in a frequency percentage difference of just over 22%.

**Table 9.** Comparison of the modes between structural-plan-based (design) FE model versus modes of OMA.

| Structural-Plan-Based FE Model | | Operational Modal Analysis | | | |
|---|---|---|---|---|---|
| Mode | Frequency (Hz) | Mode | Frequency (Hz) | % Difference | MAC |
| 2 | 2.54 | 1 | 2.55 | 0.39 | 0.95 |
| 3 | 3.68 | 2 | 3.70 | 0.54 | 0.97 |
| 3 | 3.68 | 3 | 4.72 | 22.03 | 0.88 |
| 5 | 6.95 | 4 | 6.76 | 2.81 | 0.92 |
| 13 | 12.65 | 5 | 11.59 | 9.15 | 0.74 |

The structural-plan-based (design) FE model and modes of OMA are contrasted in Table 10. The fine point-cloud-based FE model maintains a difference between the frequencies provided by the OMA data of 15% or less. The results have less of an outlier effect, even though the structural plans can obtain smaller percentage differences overall thanks to the fine point cloud's ability to obtain a smaller range. The highest MAC value is not associated with the smallest percentage difference, and the lowest MAC value is not associated with the largest percentage difference.

**Table 10.** Comparison of the modes between fine point-cloud-based FE model versus modes of OMA.

| Fine Point-Cloud-Based FE Model | | Operational Modal Analysis | | | |
|---|---|---|---|---|---|
| Mode | Frequency (Hz) | Mode | Frequency (Hz) | % Difference | MAC |
| 2 | 2.58 | 1 | 2.55 | 1.18 | 0.95 |
| 3 | 4.01 | 2 | 3.70 | 8.38 | 0.98 |
| 3 | 4.01 | 3 | 4.72 | 15.04 | 0.88 |
| 5 | 7.13 | 4 | 6.76 | 5.47 | 0.92 |
| 18 | 13.18 | 5 | 11.59 | 13.72 | 0.74 |

Table 11 contrasts the modes of the OMA modes and the coarse point-cloud-based FE model. In this instance, the percentage difference increases to 21.4%. As anticipated, compared to the coarse point cloud case, the fine point-cloud-based FE model produces the lowest worst-case percentage error and the highest overall accuracy.

**Table 11.** Comparison of the modes between coarse point-cloud-based FE model versus modes of OMA.

| Coarse Point-Cloud-Based FE Model | | Operational Modal Analysis | | | |
|---|---|---|---|---|---|
| Mode | Frequency (Hz) | Mode | Frequency (Hz) | % Difference | MAC |
| 2 | 2.60 | 1 | 2.55 | 1.96 | 0.95 |
| 3 | 3.71 | 2 | 3.70 | 0.27 | 0.97 |
| 3 | 3.71 | 3 | 4.72 | 21.40 | 0.88 |
| 5 | 7.29 | 4 | 6.76 | 7.84 | 0.92 |
| 18 | 13.70 | 5 | 11.59 | 18.21 | 0.74 |

Figure 7 shows the mode shapes and natural frequencies of the OMA, structural-plan-based (design) FE model, and fine and coarse point-cloud-based FE models. The next section presents a detailed discussion of the results of the study.

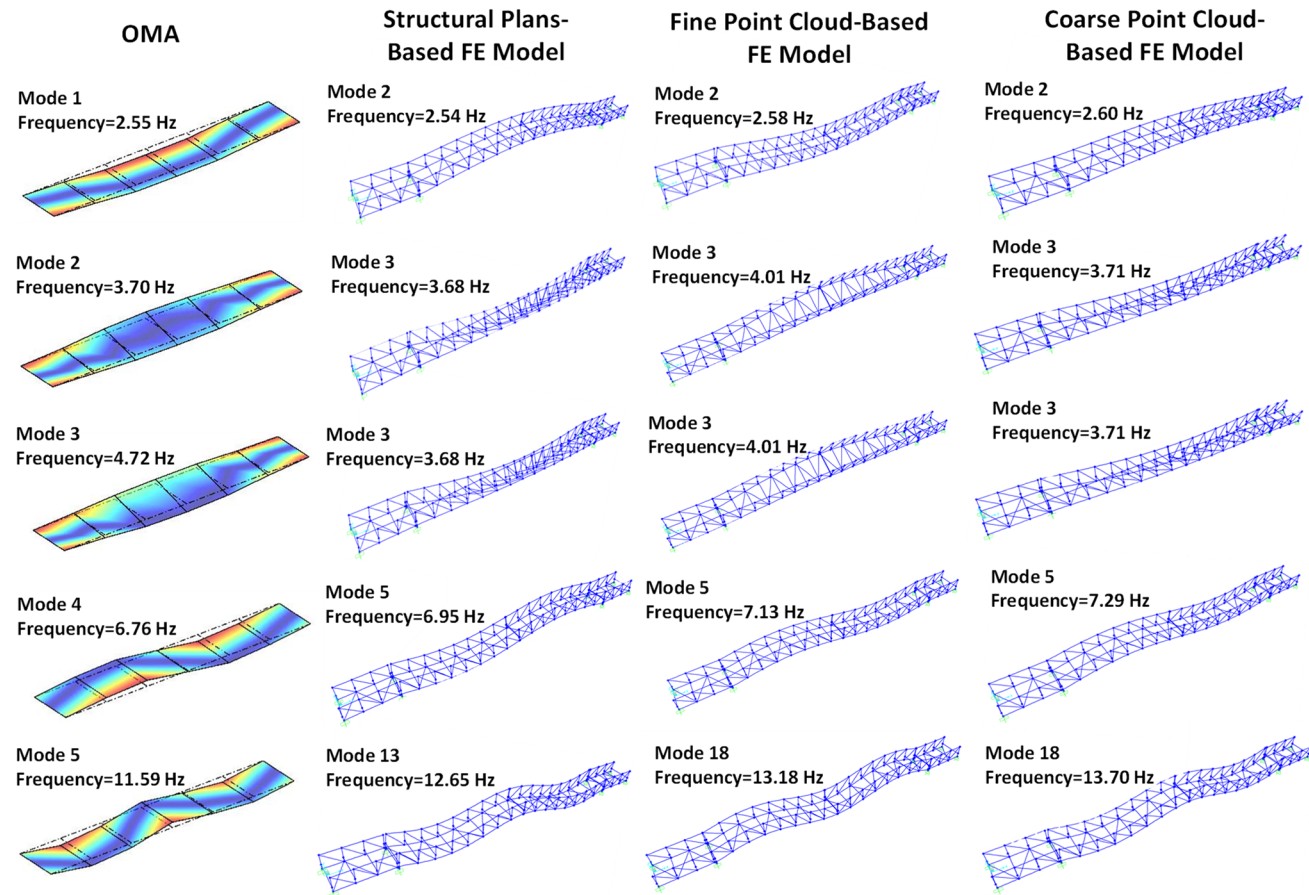

**Figure 7.** Mode shapes and natural frequencies of the modes of OMA, structural-plan-based (design) FE model, and fine and coarse point-cloud-based FE models.

It is seen that the OMA dynamic results obtained from the field data show an average 6.98% frequency difference and an average 0.89 MAC correlation with the structural-plan-based (design) FE model for the first five modes. Similarly, the OMA results show an average 8.76% and 9.94% frequency difference and an average 0.89 MAC correlation with the fine point-cloud- and coarse point-cloud-based models, respectively. It should be noted that these models are not calibrated. As such, it can be concluded that numerical models developed using point cloud data for structures with incomplete documentation and plans give comparable results.

Tables 12 and 13 overview the numeric static and dynamic and experimental analyses conducted in this study. Note that in the tables, the difference values presented throughout Tables 1–7 and Tables 9–11 are averaged and then presented in Tables 12 and 13.

**Table 12.** Summary of the results of the numeric static and dynamic analyses (FEA).

| Feature | Percentage Difference (%) | | Reference Table |
|---|---|---|---|
| | Structural-Plan-Based FE Model vs. Fine Point-Cloud-Based FE Model | Structural-Plan-Based FE Model vs. Coarse Point-Cloud-Based FE Model | |
| Midpoint displacements | 3.70 | 5.50 | Table 1 |
| Joint reactions (average of differences) | 0.075 | 0.055 | Table 2 |
| Member sizes (average of differences) | 11.76 | 4.79 | Tables 3 and 4 |

**Table 12.** *Cont.*

| Feature | Percentage Difference (%) | | Reference Table |
| | Structural-Plan-Based FE Model vs. Fine Point-Cloud-Based FE Model | Structural-Plan-Based FE Model vs. Coarse Point-Cloud-Based FE Model | |
| --- | --- | --- | --- |
| Natural frequencies—numeric (average of differences) | 4.17 | 1.75 | Table 5 |
| Load participation—static and dynamic (average of differences) | 3.88 | 6.65 | Tables 6 and 7 |

**Table 13.** Summary of the results of the experimental analyses (OMA).

| Feature | Percentage Difference (%) | | | Reference Table |
| | Structural-Plan-Based FE Model vs. Operational Modal Analysis | Fine Point-Cloud-Based FE Model vs. Operational Modal Analysis | Coarse Point-Cloud-Based FE Model vs. Operational Modal Analysis | |
| --- | --- | --- | --- | --- |
| Natural frequencies (average of differences) | 6.98 | 8.76 | 9.94 | Tables 9–11 |
| MAC values (average of differences) | 0.89 | 0.89 | 0.89 | |

## 6. Observations and Summary of the Results

This study primarily explores the feasibility of constructing numerical models for structures based on incomplete documentation and plans. These models are anticipated to be pivotal in structural preservation and monitoring data. Here, the development of models from LiDAR point cloud scans is discussed. Two point clouds are considered, one with 11 locations (scans) defined as fine and another with four locations (scans) defined as coarse point clouds.

- It can be concluded that using a point cloud is a practical technology to generate data when structural documents and plans are not available for structures. This is also a commonly encountered condition for many structures. Given that structural plans are not always accurate in terms of the dimensions, or if the condition of the structures has changed as a result of external stressors, LiDAR scans can save time compared to detailed hand measurements and also provide in situ measured geometry. Furthermore, compared to rendering from a blank canvas, as is the case for structural plans, the ability to input the data points into a program with model rendering capabilities, such as Autodesk Inventor, gives the user the base for recreating the structure and also transferring the model into digital twin platforms.
- The user's experience can have a direct impact on the outcomes. Compared to someone who has rendered point clouds before, a first-time user will find it difficult to gather results quickly and accurately. The density of the point cloud, visual obstructions of structural members, the weather, program of choice, and human judgment are just a few of the variables that affect overall accuracy. As mentioned throughout this paper, human judgment is the main contributing factor to the problem. One user can judge a member as several inches larger or smaller than another, directly impacting the outcomes generated by the FEA.
- After spending five hours on site, the first attempt at data collection had to be abandoned halfway through the scanning of the bridge due to technical issues. After spending an additional 5 hours and 45 minutes on site, the second attempt was completed. Project scheduling may suffer as a result of significant delays. However, scanning a structure thoroughly takes much less time and is more practical and

more efficient than measuring and inspecting the entire structure using conventional measuring techniques, so this aspect of using a point cloud could be advantageous.

- Due to an underlying bias that existed for the user during the rendering process, the coarse point-cloud-based FE model produces slightly more accuracy for the dynamic analysis comparison results than the structural-plan-based (design) FE model. The program operator may have fallen back on the member size thicknesses specified in the structural plans despite knowing the correct member sizes and having difficulty selecting an accurate member size due to the lack of density provided by the point cloud. Unwittingly, the program operator makes educated guesses based on the point cloud's shapes, but the known member sizes influence them. In the future, removing this bias from the study's methodology would allow it to more accurately reflect the precision of a low-density point cloud. It is also indicated that with properly identified scan locations, it is possible to achieve comparable results with respect to additional locations. In this case, four strategically identified locations saved time for data collection processing while achieving comparable results to 11 location scans. Therefore, the geometry obtained from a coarse point cloud yielded better cross-sectional member identification than the structural plans. This can be called over-fitting artifacts.

- The fine and coarse point-cloud-based FE models provide displacements of 3.7% and 3.47%, respectively, compared to the structural-plan-based (design) FE model. For the reaction comparison, the results are also similar in both cases.

- As indicated, changing the density of the point cloud does not significantly affect the overall accuracy (a coarse point cloud is slightly more accurate) of the numeric dynamic analysis results, provided that from some well-defined locations, the structure can be scanned with a clear field of view with minimum interference. As such, no natural frequency of the point-cloud-based FE model differs by more than 8.23% from another at any matching mode compared to the structural-plan-based FE model. The average frequency error concerning the design model is 4.17% and 1.75% for the fine and coarse point cloud data, respectively. As determined by human judgment and over-fitting artifacts, the different member sizes directly impact the outcomes.

- The three different numerical models' dynamic responses are compared to the operational modal analysis results of ambient vibration tests. The highest difference in frequency and the lowest MAC values between the OMA results and the dynamic analysis results of the fine point-cloud-based FE model are 15.04% and 0.74%, respectively. The largest difference in frequency and the lowest MAC values between the OMA and coarse point-cloud-based FE model are 21.40% and 0.74%, respectively. The highest frequency and lowest MAC values between the OMA and the structural-plan-based (design) FE model are 22.03% and 0.74%, respectively. The OMA results from the field data reveal a 6.98% average frequency difference and an average 0.89 MAC correlation with the structural-plan-based-FE models for the initial five modes. Similarly, when compared with fine point-cloud- and coarse point-cloud-based models, the OMA results exhibit an average frequency difference of 8.76% and 9.94%, along with an average MAC correlation of 0.89. It can be concluded that numerical models developed using point cloud data for structures with incomplete documentation and plans yield comparable results with a model with complete information.

## 7. Conclusions

The main focus of this paper is to investigate to what extent it is possible to generate numerical models of structures with incomplete documentation and plans. In such a case, LiDAR scans can be employed to generate models. These models can also be employed for structural preservation in conjunction with monitoring data and can be used in digital twinning as well as for virtual entities for immersive visualization. As a result, the Virtual Entity dimension, an essential component of digital twins, can be achieved by generating a numerical model of the existing civil structure with incomplete documentation and plans.

LiDAR is the most accurate and dependable technique to scan and deliver a 3D model of existing civil structures compared to other approaches like RADAR, SONAR, or various camera sensors, which are investigated in this paper.

This study involves analyzing a pedestrian bridge on the UCF campus by examining its geometric, static, and dynamic characteristics using point cloud data. These data are collected by a TLS and then processed to create 3D models for use in a FEA program. A comparative study is conducted to assess the FEA results for the bridge's structural-plan-based (design) FE model and two different point-cloud-based FE models with fine (11 locations) and coarse (4 locations) point cloud densities. The fine point cloud took 5 h and 45 min, while the coarse was just 1 h and 36 min. It is also seen that strategically designed point cloud scans may provide sufficiently accurate geometry and structural response results.

The study first involves a comparison of the geometries between the structural-plan-based (design) FE model and the point-cloud-based FE models for different cloud densities to investigate a case of there being no documentation for the bridge. The digital geometry obtained from the point clouds presents results comparable to the design values for the members and cross-sections.

Once the geometries are generated, the FEA results for static and dynamic cases are compared for the structural-plan-based (design) FE model and two different point-cloud-based FE models. It is also seen and summarized with numerical results that the results are also within an acceptable range, e.g., a maximum 5.5% static deformation and 8.23% frequency difference, with the average difference being less than 5% for frequencies. It should be indicated that neither the point-cloud-based models nor the design model are calibrated. Additionally, the dynamic properties of the pedestrian bridge are compared across four data sources: the structural-plan-based (design) FE model, the fine and coarse point-cloud-based FE models, and experimental data gathered using OMA in the field using golf cart excitation. Even without calibration, this dynamic analysis shows that the fine and coarse point clouds achieve a commendable average accuracy of 8.76% and 9.94% for natural frequencies and a 0.89 modal assurance criterion value, respectively.

In essence, this study contributes to the growing body of knowledge on the applicability of point cloud data in creating digital twins of civil structures. The results emphasize the potential of LiDAR-scanned point clouds in generating accurate representations of real-world structures, enabling engineers and stakeholders to make informed decisions regarding design, analysis, and maintenance. It is shown that it is possible to generate numerical models of structures with incomplete documentation and plans. For bridges with incomplete or missing structural data, practical scans with LiDAR can generate digital data for numerical modeling, for use in digital twinning, and virtual entities for immersive visualization. Future studies will include different applications to different structures, refining algorithms for improved accuracy, addressing challenges in point cloud density, and exploring immersive visualization for civil infrastructure decision-making.

**Author Contributions:** Research initiation, F.N.C. and J.A.C.; conceptualization, F.N.C. and J.A.C.; methodology, F.N.C. and J.A.C.; validation, F.N.C., J.A.C., F.L., L.C.W. and R.M.; investigation, F.N.C., J.A.C. and F.L.; resources, F.N.C. and L.C.W.; data curation, J.A.C., L.C.W. and R.M.; writing—original draft preparation, F.N.C., J.A.C. and F.L.; writing—review and editing, J.A.C., F.N.C., F.L., L.C.W. and R.M.; supervision, F.N.C. and L.C.W.; project administration, F.N.C.; funding acquisition, F.N.C. All authors have read and agreed to the published version of the manuscript.

**Funding:** This study was supported by the National Aeronautics and Space Administration (NASA) Award No. 80NSSC20K0326 and UCF Internal Seed Funding for project MAPS—Mobile Assessment for Civil Infrastructure Preservation using SHM and BIM.

**Data Availability Statement:** Some or all of the used models, code, and detailed results are available from the authors of this paper upon request.

**Acknowledgments:** The authors thank the members of the CITRS (Civil Infrastructure Technologies for Resilience and Safety) laboratory at the University of Central Florida. The paper is mostly based on

the graduate research work of Jacob Cano. Also, the authors acknowledge the contributions of Sofia Baptista and Jacob Solomon in the compilation of this study. Additionally, the authors would like to thank Paulo Dos Santos, Samantha Weiser, and Pruthviraj Thakor for their assistance throughout the data collection and Mahta Zakaria for reviewing the material. The authors would like to thank Joe Kider of the UCF School of Modeling, Simulation, and Training for the insightful discussions.

**Conflicts of Interest:** Author Jacob Cano was employed by the company Little. The remaining authors declare that the research was conducted in the absence of any commercial or financial relationships that could be construed as a potential conflict of interest.

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
