# Peer review of "On the Generation of Digital Data and Models from Point Clouds: Application to a Pedestrian Bridge Structure"

_infrastructures, doi:10.3390/infrastructures9010006_

Round 1

Reviewer 1 Report

Comments and Suggestions for Authors

please see attached files

Author Response

File attached

Reviewer 2 Report

Comments and Suggestions for Authors

This paper presents an interesting field work subject to the new concept of digital twin technology in bridge engineering, where all the bridge dimensions can be accurately obtained from LiDAR scanning. A pedestrian bridge in the college campus is utilized for field verification, with high accuracy (i.e., deformation, frequencies) obtained from both the fine and coarse scanning, compared to the results from the plan-based FE model. The paper is well-written, but a few comments need to be addressed before the acceptance,

In the section abstract, “… achieve an average accuracy of 8.75% and 9.94% for natural frequencies …” are there any typos here, like “accuracy of 8.75% and 9.94%”? meaning relative errors of nearly 90%? If not, please clearly explain.

As illustrated in Figure 4, how is the cross-section of the pedestrian bridge scanned? If possible, please give rough altitude data for the pedestrian bridge and all the scanning location points.

It is surprising to see the coarse point cloud-based model provide a better prediction, i.e., 4 collection stations are better than the 11 stations.

It is suggested to provide a table to summarize and compare all the bridge dimensions including the cross sections, from the plan-based data, the fine scanning, and the coarse scanning.

Author Response

File attached

Round 2

Reviewer 2 Report

Comments and Suggestions for Authors

All comments have been addressed by the authors. The manuscript can be accepted in its current form.